# Discriminative stimuli are sufficient for incubation of cocaine craving

**Rajtarun Madangopal[1], Brendan J Tunstall[2], Lauren E Komer[1†], Sophia J Weber[1], Jennifer K Hoots[3‡], Veronica A Lennon[1], Jennifer M Bossert[3], David H Epstein[4], Yavin Shaham[3], Bruce T Hope[1]\***

[1]Neuronal Ensembles in Addiction Section, Intramural Research Program, National Institute on Drug Abuse, National Institutes of Health, Baltimore, United States; [2]Neurobiology of Addiction Section, Intramural Research Program, National Institute on Drug Abuse, National Institutes of Health, Baltimore, United States; [3]Neurobiology of Relapse Section, Intramural Research Program, National Institute on Drug Abuse, National Institutes of Health, Baltimore, United States; [4]Real-world Assessment, Prediction, and Treatment Unit, Intramural Research Program, National Institute on Drug Abuse, National Institutes of Health, Baltimore, United States

**Abstract** In abstinent drug addicts, cues formerly associated with drug-taking experiences gain relapse-inducing potency ('*incubate*') over time. Animal models of incubation may help develop treatments to prevent relapse, but these models have ubiquitously focused on the role of conditioned stimuli (CSs) signaling drug delivery. Discriminative stimuli (DSs) are unique in that they exert stimulus-control over both drug taking and drug seeking behavior and are difficult to extinguish. For this reason, incubation of the excitatory effects of DSs that signal drug availability, not yet examined in preclinical studies, could be relevant to relapse prevention. We trained rats to self-administer cocaine (or palatable food) under DS control, then investigated DS-controlled incubation of craving, in the absence of drug-paired CSs. DS-controlled cocaine (but not palatable food) seeking incubated over 60 days of abstinence and persisted up to 300 days. Understanding the neural mechanisms of this DS-controlled incubation holds promise for drug relapse treatments.
DOI: https://doi.org/10.7554/eLife.44427.001

**\*For correspondence:**
bhope@intra.nida.nih.gov

**Present address:** †Graduate School of Medical Sciences, Weill Cornell Medicine, New York, United States; ‡Department of Psychology, University of Illinois at Chicago, Chicago, United States

**Competing interests:** The authors declare that no competing interests exist.

## Introduction

The risk of relapse is a major obstacle for effective treatment of drug addiction (*O'Brien, 2005*; *Wikler, 1973*). In abstinent drug users, several factors contribute to drug relapse, including exposure to cues and contexts previously associated with drug use (*O'Brien et al., 1992*), stressors (*Sinha, 2001*), or acute exposure to the drug itself (*Jaffe et al., 1989*). Preclinical studies have recapitulated these effects in relapse models using mice, rats, and nonhuman primates (*Venniro et al., 2016*; *Weiss, 2010*). A major finding across these studies is that cue-induced drug-seeking (in the absence of the drug) increases progressively during abstinence, a phenomenon termed incubation of drug craving (*Grimm et al., 2001*; *Neisewander et al., 2000*). Time-dependent increases in drug-seeking have been demonstrated in cocaine (*Lu et al., 2004a*), heroin (*Shalev et al., 2001*), methamphetamine (*Shepard et al., 2004*), alcohol (*Bienkowski et al., 2004*), and nicotine (*Abdolahi et al., 2010*), as well as non-drug rewards such as sucrose (*Grimm et al., 2002*). These findings in rodents mirror incubation of cue-induced drug craving and physiological responses in human addicts (*Bedi et al., 2011*; *Li et al., 2015a*; *Wang et al., 2013*; *Parvaz et al., 2016*), and have been important for studying neural mechanisms contributing to drug relapse (*Dong et al., 2017*; *Marchant et al., 2013*; *Pickens et al., 2011*; *Wolf, 2016*).

**eLife digest** More than 85% of people who give up an addictive drug begin using it again within a year. Relapse rates have changed little over the past five decades. Situations, places and objects associated with drug-taking can trigger relapse long after a person's last exposure to a drug.

We can study relapse by training animals to self-administer drugs such as cocaine. For example, rats can learn to press a lever to receive an infusion of a drug paired with a cue (a conditioned stimulus), such as a tone or a light. After training, the rats continue to press the lever to seek the drug, even if this behavior no longer delivers it. In addition, their lever pressing in response to cues increases with time for several months after their last drug exposure. This phenomenon, known as incubation of drug craving, mirrors the increase in cravings reported by abstinent drug users.

In drug users, cues such as the crack pipe or syringe used to take the drug can later contribute to drug relapse during abstinence. Most studies modeling this phenomenon have focused on how the rats respond to a conditioned stimulus that signaled the delivery of a drug during training. However, a second type of signal, known as the discriminative stimulus, can also influence relapse. Discriminative stimuli are sets of cues that signal whether drugs are about to become available or not; for example, the presence of people selling drugs on a street corner as opposed to the presence of police.

Madangopal et al. now show that discriminative stimuli – in the absence of conditioned stimuli – can also control the incubation of drug craving. Rats learned to press a lever in response to a light signaling the availability of cocaine (the positive discriminative stimulus), and to avoid responding to a different light indicating that cocaine was unavailable (the negative discriminative stimulus). When tested during abstinence, the rats only increased their lever pressing to the first light over time, i.e., they showed an incubation of drug craving controlled by the positive discriminative stimulus. Lever pressing peaked after 60 days of abstinence and persisted for up to 300 days (almost half the rats' lifespan). By contrast, the same discriminative stimuli did not trigger increased lever pressing when used to signal the availability of a palatable food.

Discriminative stimuli are thus powerful and persistent triggers of craving for addictive drugs. They signal the availability of a drug prior to both drug-taking and relapse, making them a critical target for intervention strategies. Understanding the mechanisms by which discriminative stimuli promote drug craving could lead to new treatments to prevent relapse.

DOI: https://doi.org/10.7554/eLife.44427.002

Preclinical incubation models have shown how cues presented after performance of a drug-taking response and paired with subsequent drug delivery during training potentiate drug-seeking when presented response-contingently during abstinence. These 'confirmatory' conditioned stimuli (CSs) inform the laboratory animal that the drug-taking response has been completed during training. Early preclinical studies of incubation showed that it could also occur in the absence of such discrete drug-paired CSs (*Lu et al., 2004a*; *Grimm et al., 2002*). This suggests that incubation could also be induced by other stimuli associated with drug-taking, such as the contextual cues (e.g. the chamber used for operant training) or discriminative stimuli (DSs) that signal drug availability (e.g. the house-light that is illuminated during the training session, the retractable lever that serves as the operant manipulandum). Surprisingly, little is known about the factors underlying incubation in the absence of previously drug-paired CSs. A recent study suggested that it is not mediated by contextual cues (*Adhikary et al., 2017*), leaving DSs as a likely culprit. DSs are different from cues typically investigated in these studies in that they are neither response-contingent like CSs, nor ever-present like contextual cues. Rather, DSs signal drug availability—or unavailability—thereby preceding and guiding the performance of drug-taking behavior. Previous studies have shown that a DS signaling drug availability (DS+) can promote persistent drug-seeking behavior while a DS signaling drug unavailability (DS-) can inhibit drug-taking behavior and drug-priming-induced reinstatement of drug seeking (*Weiss, 2010*; *Ettenberg, 1990*; *Gutman et al., 2017*; *Katner et al., 1999*; *McFarland and Ettenberg, 1997*; *Mihindou et al., 2013*; *Yun and Fields, 2003*; *Pitchers et al., 2017*). Further, DS control of drug seeking persists for many months and is highly resistant to extinction

(*Ciccocioppo et al., 2004*; *Ghitza et al., 2003*; *Martin-Fardon and Weiss, 2017*). Despite the importance of DSs in stimulus control of drug taking and relapse, it is unknown whether DS-controlled drug-seeking incubates during abstinence.

In this study, we sought to directly assess the contribution of DSs to incubation, in the absence of drug-paired CSs. To this end, we first designed a trial-based procedure to train male and female rats to discriminatively self-administer cocaine (0.75 mg/kg/infusion) during trials in which a DS+ signaled cocaine availability, and to suppress responding on the same lever during trials in which a DS- signaled cocaine unavailability during the same session. Drug infusions were not paired with CSs. We then tested for the ability of DSs to control cocaine seeking at multiple time points extending up to 400 days of abstinence. Further, after complete cessation of cocaine-seeking behavior, we assessed whether a priming dose of cocaine would reinstate DS-controlled cocaine seeking in the same rats. Finally, to determine whether DS-controlled incubation under our experimental conditions was specific to cocaine, we trained a separate group of rats on an analogous procedure using palatable food (45 mg high-carbohydrate pellets) as the operant reward and assessed the time course of DS-controlled food seeking.

## Results

### Experiment 1: incubation of discriminative stimulus-controlled cocaine seeking

The goal of Experiment 1 was to determine the persistence of non-reinforced discriminated cocaine seeking (*relapse* to DS-controlled cocaine seeking) and to test for the potentiation of this seeking response during abstinence (*incubation* of DS-controlled cocaine seeking). We trained male and female rats to press a central retractable lever only during trials in which lever entry was preceded by the illumination of a light stimulus that signaled cocaine availability (DS+ trials) and to suppress responding during trials when availability of the same lever was preceded by a second light stimulus signaling absence of cocaine reward (DS- trials). There were no additional reward-paired discrete cues. Once trained, we used a within-subjects design to test for discriminated cocaine seeking (extinction conditions) after varying durations of abstinence extending up to 400 days. After complete cessation of cocaine-seeking behavior on abstinence day 400, we used a within-subjects design and an ascending cocaine dose-response procedure to assess the ability of priming injections of cocaine to reinstate DS-controlled cocaine-seeking. All behavioral data pertaining to Experiment 1 are shown in *Figure 1* (collapsed across sex), and *Figure 1—figure supplement 1* (disaggregated by sex). Statistical outputs for all analyses pertaining to the experiment are provided in tabular format as *Figure 1—source data 1*.

#### Training

The experimental timeline and individual trial design are shown in *Figure 1A,B*. Rats learned to respond on the lever for cocaine reward during the first six sessions of continuous access (*Figure 1C*). They continued responding in the trial format and then learned to discriminate DS + from DS- during discrimination training. The number of 'successful' trials (denoted as *trials* and defined as making at least one lever press during a trial) and total number of lever presses (denoted as *lever presses* and recorded separately for each DS trial type) during each session were analyzed. We used a two-way maximum-likelihood-based multilevel model with within-subjects factors Session (discrimination sessions 5–14) and DS (DS+, DS-). For *trials*, we observed a significant main effect of DS ($F_{1,13}$=948.21, p<0.0001) but not of Session, and no interaction between Session and DS, indicating that responding during DS+ trials was higher than responding during DS- trials during all the discrimination training sessions. For *lever presses* , we observed a significant main effect of DS ($F_{1,13}$= 161.63, p<0.0001) and an interaction between Session and DS ($F_{9,117}$=2.62, p=0.0085) but no main effect of Session. Post-hoc analyses indicated that responding during DS+ trials was higher than responding in DS- trials during the last four discrimination training sessions (p<0.05).

#### Relapse test

*Figure 1D* shows relapse in terms of mean responding during 3 hr non-reinforced discrimination test sessions for cocaine seeking. The same rats were tested at different time points 1–400 days

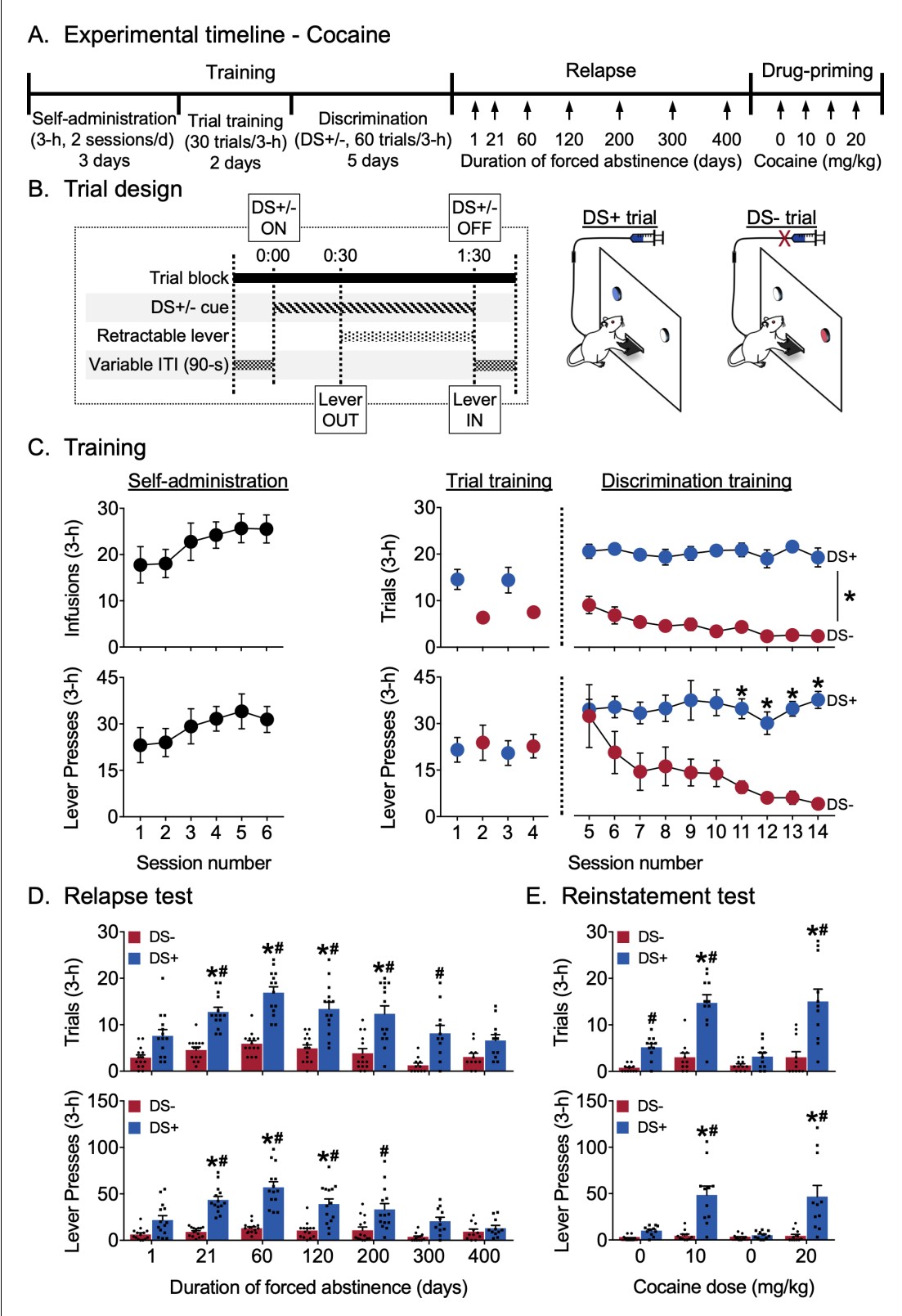

**Figure 1.** Incubation of discriminative stimulus-controlled cocaine seeking. (**A**) Experimental timeline. (**B**) Schematic showing the timing of individual events during a single 3 min DS trial, and the differences between the two trial types during discrimination training for cocaine reward. Rats received cocaine reward (0.75 mg/kg/infusion) during DS+ trials but did not receive cocaine reward during DS- trials (n = 16). (**C**) Training data. *Self-administration*: Rats learned to self-administer cocaine over six sessions. Mean (±SEM) number of cocaine infusions and lever presses during each 3 hr

*Figure 1 continued on next page*

*Figure 1 continued*

session. *Trial training:* Mean (±SEM) number of DS+ or DS- trials with at least one lever press (denoted as *trials*), and number of lever presses during the 3 hr sessions (denoted as *lever presses*) with 30 trials of a single-trial type (DS+ trials in the AM session, DS- trials in the PM session). *Discrimination training:* Over 10 sessions, rats learned to discriminate DS+ from DS- trials. Mean (±SEM) number of *trials* and *lever presses* during the 3 hr discrimination training session with 30 trials each of DS+ and DS- trials presented in a pseudorandomized manner. *indicates significant difference (p<0.05) between responding during DS+ and DS- trial types (n = 14). (D) Relapse test. Incubation of lever responding during DS+, but not DS-, trials peaked at 60 days of abstinence and returned to basal levels over 400 days. Mean (±SEM) number of *trials* and *lever presses* during the 3 hr relapse test sessions (30 trials each of DS+ and DS- presented in a pseudorandomized manner) under extinction conditions. *denotes significant (p<0.05) difference from responding during day 1. Columns indicate mean (±SEM) for the group, while dots indicate values for individual rats. #denotes significant (p<0.05) difference between DS+ and DS- responding during the test sessions (n = 11–14). (E) Reinstatement test. Rats reinstated DS-controlled cocaine-seeking in response to IP injections of cocaine (10 and 20 mg/kg), but not saline. Mean (±SEM) number of *trials* and *lever presses* during the 3 hr saline- or cocaine-primed reinstatement test sessions (30 trials each of DS+ and DS- presented in a pseudorandomized manner) under extinction conditions. Columns indicate mean (±SEM) for the group, while dots indicate values for individual rats. *denotes significant (p<0.05) difference from responding on the first saline-prime test session (cocaine dose = 0 mg/kg). #denotes significant difference (p<0.05) between DS+ and DS- responding during the test session (n = 11). See *Figure 1—figure supplement 1* for behavioral data from the experiment, disaggregated by sex. See *Figure 1—figure supplement 2* for subject body weights, disaggregated by sex. See *Figure 1—source data 1* for a table of statistical output relating to the experiment.

DOI: https://doi.org/10.7554/eLife.44427.003

The following source data and figure supplements are available for figure 1:

**Source data 1.** Statistical output for Experiment 1: Incubation of discriminative stimulus-controlled cocaine seeking (analyses pertaining to *Figure 1* are highlighted in grey).
DOI: https://doi.org/10.7554/eLife.44427.006
**Figure supplement 1.** Behavioral data disaggregated by sex.
DOI: https://doi.org/10.7554/eLife.44427.004
**Figure supplement 2.** Subject body weights disaggregated by sex.
DOI: https://doi.org/10.7554/eLife.44427.005

following discrimination training. As during training, we analyzed both *trials* and *lever presses* measures. We used a two-way factorial model with within-subjects factors of Days of forced abstinence (1, 21, 60, 120, 200, 300, and 400 days) and DS type (DS+, DS-). For *trials*, we observed significant main effects of Day ($F_{6,72}$=9.68, p<0.0001) and DS ($F_{1,13}$=257.53, p<0.0001) and an interaction between the two ($F_{6,72}$=4.30, p=0.0009), reflecting higher numbers of 'successful' trials associated with DS+ presentation after 21, 60, 120 and 200 abstinence days compared to that at one abstinence day (p<0.05) and more 'successful' DS+ trials compared to DS- trials on abstinence days 21, 60, 120, 200 and 300 (p<0.05). The number of 'successful' DS- trials did not significantly increase over days. For *lever presses*, we observed significant main effects of Day ($F_{6,72}$=8.94, p<0.0001) and DS ($F_{1,13}$=182.25, p<0.0001), and an interaction between the two factors ($F_{6,72}$=7.95, p<0.0001), reflecting higher numbers of lever presses associated with DS+ presentation after 21, 60, and 120 abstinence days compared to that at one abstinence day (p<0.05) and higher responding during DS + trials compared to DS- trials on abstinence days 21, 60, 120 and 200 (p<0.05). The number of lever presses during DS- trials did not significantly increase over days. Overall, the *trial* data indicate incubation of 'successful' trials during DS+, but not DS-, trials after 21–200 days of abstinence, while the *lever presses* data indicate incubation of *the* number of lever presses during DS+, but not DS-, trials after 21–120 days abstinence. Further, the rats maintained discriminative responding up to 300 days (by the trials measure) or 200 days (by the lever presses measure) after the last training session.

## Reinstatement test

*Figure 1E* shows reinstatement in terms of mean responding during a 3 hr non-reinforced discrimination session for cocaine seeking after priming injections of either cocaine or saline. For both *trials* and *lever presses* measures, we used a two-way factorial model with the within-subjects factors Treatment condition (Saline 1, 10 mg/kg cocaine, Saline 2, 20 mg/kg cocaine) and DS type (DS+, DS-). For *trials*, there were significant main effects of Treatment ($F_{3,30}$=15.35, p<0.0001) and DS ($F_{1,10}$=108.66, p<0.0001) and an interaction between the two factors ($F_{3,30}$=12.42, p<0.0001), reflecting higher numbers of 'successful' trials associated with DS+ presentation after both cocaine priming doses (10 and 20 mg/kg) compared to saline and higher responding during DS+ trials compared to DS- trials following both cocaine priming doses as well as after the first saline priming

injection (p<0.05). The number of 'successful' trials associated with DS- presentation was not altered by the Treatment conditions. For *lever presses*, we observed significant main effects of Treatment ($F_{3,30}$=8.31, p=0.0004) and DS ($F_{1,10}$=45.73, p<0.0001) and an interaction between the two factors ($F_{3,30}$=9.45, p=0.0001), reflecting higher numbers of lever presses during DS+ trials after both cocaine-priming injections (10 and 20 mg/kg) compared to saline (Saline1 and Saline2), as well as higher responding during DS+ trials compared to DS- trials during both cocaine-priming injections (p<0.05). The number of lever presses associated with DS- presentation was not altered by the Treatment conditions. Overall, the *trials* and *lever presses* data indicated reliable cocaine-primed reinstatement during DS+, but not DS-, trials that occurred more than 400 days after the last discrimination training session.

## Experiment 2: abatement of discriminative stimulus-controlled palatable food-seeking

The goal of Experiment 2 was to determine whether the persistence and potentiation of DS-controlled seeking observed in Experiment one would generalize to a palatable food reward. We trained male and female rats to lever press for palatable food reward using a training procedure similar to that in Experiment 1. Following training, we used a within-subjects design to test the rats for discriminated palatable food-seeking after varying durations of abstinence extending up to 200 days. All behavioral data pertaining to Experiment 2 are shown in *Figure 2* (collapsed across sex), and *Figure 2—figure supplement 1* (disaggregated by sex). Statistical outputs for all analyses pertaining to the experiment are provided in tabular format as *Figure 2—source data 1*.

### Training

Rats learned to respond on the lever for palatable food reward during the first three continuous access sessions (*Figure 2C*). The rats continued responding in the trial format and then learned to discriminate DS+ from DS- during discrimination training. For analysis of successful discrimination on both *trials* and *lever presses* measures, we used a two-way factorial model with within-subjects factors Session (discrimination training sessions 3–13) and DS (DS+, DS-). For *trials*, we observed significant main effects of Session ($F_{10,140}$=15.42, p<0.0001) and DS ($F_{1,14}$=1014.94, p<0.0001) and an interaction between the two factors ($F_{10,140}$=8.22, p<0.0001), reflecting higher responding during DS+ trials for all but the first session (p<0.05). For *lever presses*, we observed significant main effects of Session ($F_{10,140}$=15.63, p<0.0001) and DS ($F_{1,14}$=577.71, p<0.0001) and an interaction between the two factors ($F_{10,140}$=2.31, p=0.0151), again reflecting higher responding during DS+ trials for all but the first session (p<0.05).

### Relapse test

*Figure 2D* shows relapse in terms of mean responding during 2 hr non-reinforced discrimination test sessions for palatable food seeking. The same rats were tested at different time points 1–200 days following discrimination training. As during training, we analyzed both *trials* and *lever presses* measures. For both *trials* and *lever presses* measures, we used a two-way factorial model with within-subjects factors of Days of forced abstinence (1, 21, 60, 120, and 200 days) and DS type (DS+, DS-). For *trials*, we observed significant main effects of Day ($F_{4,56}$=5.57, p=0.0008) and DS ($F_{1,14}$=133.04, p<0.0001) and an interaction between the two factors ($F_{4,56}$=14.29, p<0.0001), reflecting lower numbers of 'successful' trials associated with DS+ presentation after 60, 120 and 200 abstinence days than after one abstinence day (p<0.05) and higher responding to DS+ than DS- on abstinence days 1, 21, and 60 (p<0.05). The number of 'successful' trials associated with DS- presentation did not increase over days. For *lever presses*, we observed significant main effects of abstinence Day ($F_{4,56}$=8.57, p<0.0001) and DS ($F_{1,14}$=101.92, p<0.0001) and an interaction between the two ($F_{4,56}$=17.44, p<0.0001), reflecting lower numbers of lever presses associated with DS+ presentation after 21, 60, 120 and 200 abstinence days than after one abstinence day (p<0.05) and higher responding to DS+ than DS- on abstinence days 1 and 21 (p<0.05). The number of lever presses associated with DS- presentation did not change over days. Overall, *trials* and *lever presses* data indicate that food seeking decreased or abated over time and that the rats maintained discriminative responding for only 60 days (by the *trials* measure) or 21 days (by the *lever presses* measure) after the last training session.

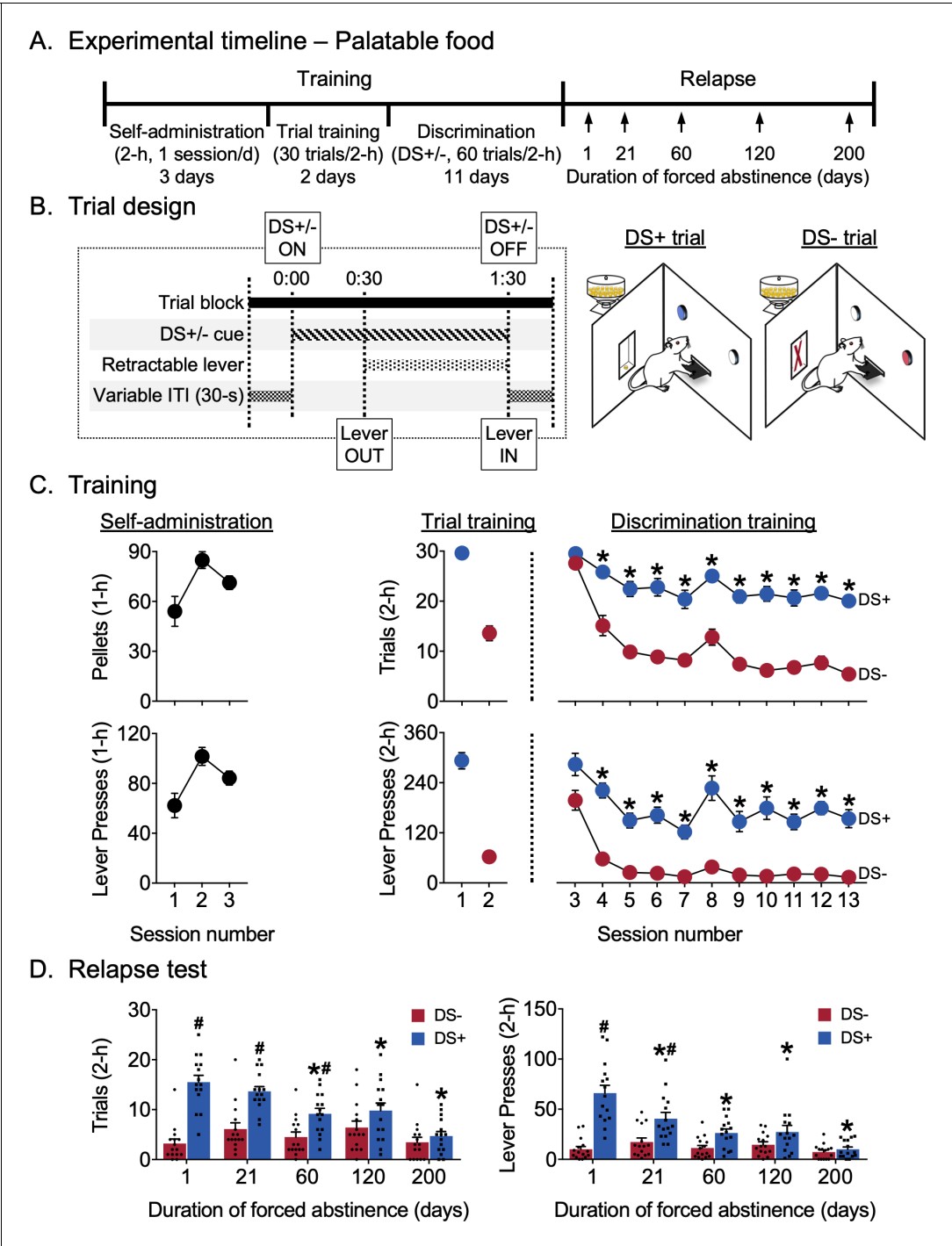

**Figure 2.** Abatement of discriminative stimulus-controlled palatable food seeking. (A) Experimental timeline. (B) Schematic showing the timing of individual events during a single 2-min DS trial, and the differences between the two trial types during discrimination training for palatable food reward (45 mg high carbohydrate pellets). Rats received food reward during DS+ trials but did not receive reward during DS- trials (n = 16). (C) Training data. *Self-administration*: Rats learned to self-administer palatable food over three sessions. Mean (±SEM) number of palatable food pellets received and lever presses during each 1 hr session. *Trial training*: Mean (±SEM) number of DS+ or DS- trials with at least one lever press (denoted as *trials*), and number of lever presses during the 2 hr sessions (denoted as *lever presses*) with 30 trials of a single trial type (DS+ trials in the AM session, DS- trials in the PM session). *Discrimination training*: Over 11 sessions, rats learned to discriminate DS+ from DS- trials. Mean (±SEM) number of *trials* and *lever presses* during the 2 hr discrimination training session with 30 trials each of DS+ and DS- trials presented in a pseudorandomized manner. *indicates significant difference (p<0.05) between responding during DS+ and DS- trials (n = 15). (D) Relapse test. Lever responding during DS+, but not DS-, trials peaked at 1 day of abstinence and abated over 200 days. Mean (±SEM) number of *trials* and *lever presses* during the 2 hr relapse test sessions (30

*Figure 2 continued on next page*

*Figure 2 continued*

trials each of DS+ and DS- presented in a pseudorandomized manner) under extinction conditions. *denotes significant (p<0.05) difference from responding during day 1. Columns indicate mean (±SEM) for the group, while dots indicate values for individual rats. #denotes significant (p<0.05) difference between DS+ and DS- responding during the test (n = 15). See *Figure 2—figure supplement 1* for behavioral data from the experiment, disaggregated by sex. See *Figure 2—figure supplement 2* for food rewards earned during discrimination training, disaggregated by sex. See *Figure 2—figure supplement 3* for subject body weights, disaggregated by sex. See *Figure 2—source data 1* for a table of statistical output relating to the experiment.

DOI: https://doi.org/10.7554/eLife.44427.007

The following source data and figure supplements are available for figure 2:

**Source data 1.** Statistical output for Experiment 2: Abatement of discriminative stimulus-controlled palatable food-seeking (analyses pertaining to *Figure 2* are highlighted in grey).

DOI: https://doi.org/10.7554/eLife.44427.011

**Figure supplement 1.** Behavioral data disaggregated by sex.

DOI: https://doi.org/10.7554/eLife.44427.008

**Figure supplement 2.** Food rewards earned disaggregated by sex.

DOI: https://doi.org/10.7554/eLife.44427.009

**Figure supplement 3.** Subject body weights disaggregated by sex.

DOI: https://doi.org/10.7554/eLife.44427.010

## Discussion

We used a trial-based procedure to investigate incubation of cocaine or palatable-food seeking controlled by discriminative stimuli that signal availability (DS+) or unavailability (DS-) of the rewards in the absence of reward-paired CSs. Rats readily learned to respond to the DS+ for either cocaine (Experiment 1) or food (Experiment 2) and to inhibit responding to the DS- within the same session. DS-controlled cocaine seeking was maximal after 60 days of abstinence (reflecting incubation of DS-controlled cocaine seeking) and persisted for up to 300 days. Additionally, when DS-controlled cocaine seeking was fully extinguished after 400 days of abstinence, priming injections of cocaine reinstated cocaine seeking. In contrast, DS-controlled food seeking was maximal at 1 day of abstinence, progressively decreased over time, and was no longer observed after 60 abstinence days. Thus, incubation of DS-controlled reward seeking under our experimental conditions was specific for cocaine.

In previous studies, DSs paired with cocaine self-administration have been shown to promote drug seeking that is highly resistant to extinction across multiple non-reinforced test sessions (*Ciccocioppo et al., 2004*; *Martin-Fardon and Weiss, 2017*; *Weiss et al., 2000*). Reward deliveries in these studies were paired with additional discrete CSs, and the contrasting DSs were paired with different levers and presented in separate sessions, making it difficult to disentangle the potential contribution of DSs from CSs and contextual stimuli. In our experiments, reward deliveries were not paired with additional CSs during training, and the two contrasting DSs were paired with a common retractable lever and presented in a pseudorandomized order within the same session. Following training, the rats were tested under non-reinforced conditions for DS-controlled drug seeking, using the same DS presentation schedule as during training. Because the same operant manipulandum and response was required to seek reinforcement in response to each DS within the same test session, we know that discriminated drug seeking in our model was exclusively controlled by the DSs and not by contextual stimuli, classically conditioned spatial cues, presentation of the operant manipulandum, or even performance of the drug-seeking response. Under these conditions, we observed persistent non-reinforced drug seeking during DS+ presentations but not DS- presentations, up to 300 days after the last DS-drug pairing. These data extend previous studies of DS-controlled drug-seeking and suggest that in addition to setting the occasion for drug-seeking behavior, the DS+ is sufficient to motivate drug-seeking in the absence of explicit drug-paired CSs.

It has long been appreciated in basic behavioral research that learning about operant DSs involves both classical and operant conditioning (*Mowrer, 1960*; *Rescorla and Solomon, 1967*; *Weiss, 1978*; *Weiss, 2014*). As explained by *Rescorla and Solomon (1967)*, all conditions necessary for classical conditioning are present during discriminated operant responding. Thus, in addition to setting the occasion for reward-taking it should be expected that a DS+ will come to elicit classically conditioned responses (CRs). Such CRs can include the induction of motivational states (i.e. drug

craving). In the present study, rats learned to lever press during the DS+ for cocaine (i.e. they learned a stimulus-response-outcome relation and their behavior came under stimulus control). Because they received cocaine only during the DS+, they should have also learned a Pavlovian stimulus-outcome relation that would imbue the DS+ with incentive motivational properties (*Weiss, 2014*). These excitatory motivational properties, acquired through Pavlovian processes, likely contributed to the incubation effect observed in this study.

We demonstrate that DS-controlled cocaine seeking is potentiated during abstinence (that is, we show incubation of DS-controlled cocaine craving) even in the absence of explicit drug-paired CSs. Incubation studies have typically employed between-subjects testing procedures in which rats previously trained to self-administer an addictive drug are returned to the same chambers after varying periods of abstinence and tested for drug-seeking with or without the previously drug-paired CSs (*Pickens et al., 2011*; *Li et al., 2015b*; *Lu et al., 2004b*). Following cocaine self-administration, the response to cocaine-paired CSs progressively increased ('incubated') over the first 60–90 days of withdrawal (*Grimm et al., 2001*; *Grimm et al., 2003*). However, incubation has also been observed in the absence of drug-paired CSs; extinction responding in the absence of the drug-paired CSs also progressively increased for up to 90 days (*Grimm et al., 2003*) and persisted up to 180 days (*Lu et al., 2004a*). A recent study demonstrated that this potentiation is not mediated by contextual cues (*Adhikary et al., 2017*). However, the factors controlling incubation in the absence of previously drug-paired CSs were not elucidated. In the present study, we found a time-dependent increase in drug seeking (incubation) during DS+ presentations in the absence of any explicit drug-paired CSs during abstinence. DS-controlled seeking continued to increase up to 60 days into abstinence and persist up to 300 days (about half the lifespan of a rat). This time course of DS-controlled cocaine-seeking is especially remarkable when considering that the same group of rats were exposed to repeated relapse tests under extinction conditions. It is possible that in a between-subjects design, DS-controlled incubation would show a longer rise phase than the one observed here and persist beyond 300 days, in the absence of extinction learning over repeated relapse tests.

In contrast, cocaine seeking in DS- trials did not incubate – rats continued to suppress responding in DS- trials during all relapse tests and maintained discrimination up to 300 days of abstinence. DSs signaling cocaine unavailability have been shown to inhibit ongoing cocaine self-administration and to suppress cocaine priming-induced reinstatement (*Mihindou et al., 2013*). From the perspective of translation and treatment development, the inhibition of cocaine seeking may be just as important as its potentiation. The behavior guided by each DS in our study was able to survive multiple extinction sessions and subsequent tests of cocaine priming-induced reinstatement – after DS+ responding was extinguished to DS- levels – priming injections of cocaine reinstated cocaine seeking specifically during DS+, but not DS-, trials. Future studies with this procedure will dissociate the neurobiological mechanisms that allow these two functionally orthogonal DSs to mediate incubation of DS-controlled cocaine seeking.

Using a similar format of DS and lever presentation, we also trained rats to lever press for palatable food reward during DS+, but not DS-, trials. We found that food-DS rats made more total responses than cocaine-DS rats during training, maintained their discrimination responding under non-reinforced conditions, and also showed higher seeking responses than cocaine-DS rats during the initial relapse test on day 1. However, under the same repeated-testing schedule used for cocaine relapse, they quickly extinguished their DS-controlled responding in the absence of food and progressively decreased food seeking during abstinence. It is possible that DS-controlled food seeking would have incubated in the absence of repeated relapse testing in extinction. Indeed, incubation has been observed using the classical procedure with oral sucrose reward; sucrose reward-seeking during presentation of the previously reward-paired CSs (cue-induced reinstatement) progressively increased during abstinence, peaked at 30 days and abated at 90 days of abstinence (*Lu et al., 2004b*; *Grimm et al., 2003*). It is unlikely that the differences in non-drug seeking in response to CSs and DSs are the result of the choice of non-drug reinforcer as a recent study demonstrated incubation of CS-induced reward seeking using the same palatable food pellets (*Krasnova et al., 2014*). The greater persistence of seeking in response to drug- over food-DSs observed here is more likely a result of an inherent difference in the strength of stimulus-control exerted by drug- over food-DSs. Our findings are in agreement with earlier studies directly comparing drug and food paired-DSs (*Ciccocioppo et al., 2004*; *Martin-Fardon and Weiss, 2017*) but also more broadly, with studies comparing drug- and food-paired CSs (*Tunstall and Kearns, 2016*;

*N. Kearns et al., 2011*). Future studies are required to determine whether this divergence of DS effects on drug versus food seeking is due to differences in the strength of the initial DS-reward associations during training or due to drug-specific neuroadaptations that emerge during abstinence (*Wolf, 2016*; *Grimm et al., 2003*; *Shaham and Hope, 2005*).

Taken together, the results of the present experiments show that DS-controlled operant drug seeking incubates during prolonged abstinence and persists up to 300 days of abstinence despite repeated relapse testing. However, using a similar repeated testing procedure, we observed an abatement of DS-controlled palatable food seeking. As we noted above, DS-controlled behaviors offer an especially promising path to treatment development because DSs are always present before and during human drug taking; they do not merely accompany or follow it. They can play a critical role in relapse; for example, a study measuring flight attendants' cigarette craving showed that craving peaked toward the end of flights as the opportunity to smoke a cigarette neared, regardless of flight duration or time since the last cigarette (*Dar et al., 2010*). Animal models of other aspects of addictive behavior have been questioned, by us and other authors, because the timing or sequencing of events does not reflect the typical experiences of human drug users (*Epstein and Kowalczyk, 2018*; *Vanderschuren et al., 2017*). The procedure we describe here addresses those concerns in the realm of cue reactivity and its incubation, and is well suited to disentangle the complex array of behavioral and neural mechanisms underlying the contributions of DSs to relapse (*Bradfield and Balleine, 2013*; *Colwill and Rescorla, 1990*; *Rescorla, 1990*; *de Wit and Dickinson, 2009*).

## Materials and methods

### Experimental design

The goal of this study was to test for the ability of discriminative stimuli signaling cocaine availability to potentiate cocaine-seeking after withdrawal and then determine if this effect would generalize to non-drug rewards. A detailed description of experimental subjects, apparatus and procedures is included in the following subsection. We first provide an overview of the specific behavioral experiments.

### Experiment 1: incubation of discriminative stimulus-controlled cocaine-seeking

The goal of Experiment 1 was to determine the persistence of non-reinforced discriminated cocaine seeking (*relapse* to DS-controlled cocaine-seeking) and to test for the potentiation of this seeking response during abstinence (*incubation* of DS-controlled cocaine seeking). The timeline of the experiment is shown in *Figure 1A*. We trained male and female rats using two 3 hr daily sessions (morning and afternoon) to press a central retractable lever only during trials in which lever entry was preceded by the illumination of a light stimulus that signaled cocaine (0.75 mg/kg/infusion) availability (DS+ trials) and to suppress responding during trials when availability of the same lever was preceded by a second light stimulus signaling absence of cocaine reward (DS- trials). There were no additional reward-paired discrete cues. After successful training (20 sessions over 10 days), we tested for discriminated cocaine seeking using a within-subjects design, after varying durations of abstinence extending up to 400 days. For relapse testing, we used the same trial-based procedure and recorded the number of successful trials (defined as making at least one lever press during a trial) and the total number of lever presses (recorded separately for DS+ and DS- trials) made during the 3 hr sessions. Finally, after complete cessation of cocaine-seeking behavior on abstinence day 400, we assessed the ability of priming injections of cocaine to reinstate DS-controlled cocaine-seeking using a within-subjects design and an ascending cocaine dose-response procedure.

### Experiment 2: abatement of discriminative stimulus-controlled palatable food-seeking

The goal of experiment two was to determine whether the persistence and potentiation of responding seen in Experiment 1 would generalize to a nondrug reward. The timeline of the experiment is shown in *Figure 2A*. We first trained male and female rats using 2 hr daily sessions (morning or afternoon) to lever press for palatable food reward (45 mg high-carbohydrate pellets) using a procedure

similar to the one used in Experiment 1. After successful training (16 sessions over 16 days), we tested all rats for discriminated palatable food-seeking using a within-subjects design similar to that in Experiment 1 but using 2 hr sessions, after varying durations of abstinence extending up to 200 days.

## Subjects

We used male (n = 16) and female (n = 16) Sprague-Dawley rats (Charles River, USA; RRID: RGD_70508), weighing 250–350 g prior to surgery and training. In experiment 1 with cocaine self-administration training, we pair-housed rats of the same sex for 1 week (n = 8 each male and female) prior to surgery and individually housed them after intravenous surgery, during training and abstinence phases. In Experiment 2 with food self-administration training, we pair-housed rats of the same sex for 1 week (n = 8 each male and female) prior to the start of behavioral training and individually housed them during training and abstinence. For both experiments, we maintained the rats in the animal facility under a reverse 12:12 hr light/dark cycle with free access to standard laboratory chow and water in their home cages throughout the experiment. All procedures followed the guidelines outlined in the Guide for the Care and Use of Laboratory Animals (8th edition; http://grants.nih.gov/grants/olaw/Guide-for-the-Care-and-Use-of-Laboratory-Animals.pdf). In Experiment 1, 14 rats successfully completed discrimination training. We excluded one female rat due to catheter patency failure and one male rat due to failure to acquire drug self-administration. Two male rats and one female rat died during the abstinence period. In Experiment 2, all 16 rats successfully completed discrimination training. One male rat died during the abstinence period. For both experiments, we used maximum-likelihood-based multilevel models (SAS Proc Mixed) to account for missing data.

## Drugs

We received 100 mg/ml cocaine-HCl (cocaine) diluted in sterile saline from the NIDA pharmacy. We chose a unit dose of 0.75 mg/kg per infusion for self-administration training based on previous studies (Koya et al., 2009) and maintained the same unit dose during discrimination training.

## Intravenous surgery

For Experiment 1, we implanted the rats with silastic catheters in their right jugular vein using previously described methods (Adhikary et al., 2017). We anesthetized the rats with isoflurane gas (5% induction, 1–3% maintenance) and inserted silastic catheters into the jugular vein. We passed the catheters subcutaneously to the mid-scapular region and attached them to modified 22-gauge cannulae (PlasticsOne, USA) cemented in polypropylene mesh (Small Parts Inc, USA) placed under the skin. We administered ketoprofen (2.5 mg/kg, subcutaneous injection; Henry Schein Inc, USA) after surgery to relieve pain and allowed rats to recover for 5–7 days prior to drug self-administration training. We flushed the catheters daily with sterile saline containing gentamicin (4.25 mg/ml; Fresenius Kabi, USA) during the recovery and training phases. We weighed rats prior to each daily behavioral session, over the course of each experiment. Subject body weights for each experiment (disaggregated by sex) are shown as figure supplements linked to the main figures.

## Apparatus

We trained and tested all rats in standard Med Associates (Med Associates Inc, USA) self-administration chambers (Med Associates ENV-007) enclosed in a ventilated, sound-attenuating cabinet with blacked out windows. Each chamber was equipped with a stainless steel grid floor and two side-walls, each with three modular operant panels. For Experiment 1, we equipped the right-side wall of the chamber with a single retractable lever in the center panel, 7.5 cm above the grid floor. We positioned a discriminative stimulus (light, Med Associates ENV-221M) that signaled cocaine availability on the left panel and another discriminative stimulus (light, Med Associates ENV-221M) that signaled unavailability of cocaine on the right panel of the same side wall, equidistant from the central retractable lever and 14.0 cm above the grid floor. In addition to location, we used red or white lens caps to differentiate between the two discriminative cues and counterbalanced them across the 14 boxes used for Experiment 1. We connected the rat's catheter to a liquid swivel (Instech Laboratories Inc, USA) via polyethylene-50 tubing that was protected by a metal spring and used a 20-ml syringe driven by a single speed syringe pump (Med Associates PHM-100, 3.33 RPM) placed outside the

sound-attenuating cabinet to deliver intravenous cocaine infusions. In Experiment 2, we used eight different self-administration chambers. We equipped the left side wall of these chamber with a single retractable lever in the center panel, 7.5 cm above the grid floor. We positioned a discriminative stimulus (light, Med Associates ENV-221M) that signaled availability of palatable food reward on the right panel and another discriminative stimulus (light, Med Associates ENV-221M) that signaled unavailability of food reward on the left panel of the same side-wall, equidistant from the central retractable lever and 14.0 cm above the grid floor. We again used red or white lens caps to differentiate between the two discriminative cues and counterbalanced them across the boxes used for this experiment. We equipped the central panel of the opposite (right) wall with a pellet receptacle (Med Associates ENV-200R2M-6.0) connected to a 45-mg pellet dispenser (Med Associates ENV-203–45) to deliver palatable food-reward.

## Behavioral procedures

Experimental timelines for each experiment are shown in *Figures 1A* and *2A*. The self-administration, trial and discrimination training phases for cocaine and food experiments are described separately below. The subsequent abstinence and relapse test phases are the same for both experiments and described together.

### Cocaine self-administration

We trained male and female rats to lever press for cocaine reward during two 3 hr sessions per day that were separated by 30–60 min. We gave rats Froot Loops (Kellogg Company, USA) in their home cage 1 day prior to the start of training and then used crushed Froot Loops when necessary to encourage rats to engage with the lever during initial continuous access training. The start of a session was signaled by the illumination of a light cue on the right side of the retractable lever followed 30 s later by the presentation of the central retractable lever for 180 min. The light remained on for the duration of the session and served as a discriminative stimulus for cocaine reward availability. The same light was later used as a discriminative stimulus to signal availability of cocaine during trial-based discrimination training. Throughout the session, responses on this lever were rewarded under a fixed-ratio-1 (FR1) reinforcement schedule and cocaine at a unit dose of 0.75 mg/kg/infusion (0.1 ml/infusion) was delivered over 3.5 s. This infusion duration also served as the timeout period, during which lever presses were recorded but not reinforced. It is important to note that the delivery of cocaine was not paired with any discrete cues. At the end of each 3 hr session, the discriminative stimulus was turned off and the lever was retracted. We recorded (*O'Brien, 2005*) the total number of lever presses and (*Wikler, 1973*) the total number of infusions received during the entire session. We gave rats up to six training sessions to acquire stable self-administration responding in the continuous access procedure before switching them to trial training for cocaine reward.

### Trial training for cocaine reward

We trained rats in two 3-hr trial training sessions per day for 2 days. We gave rats trial training sessions before trial-based discrimination training to (*O'Brien, 2005*) habituate rats to the trial format and (*Wikler, 1973*) introduce the two possible trial contingencies separately before we mixed them together during discrimination sessions. The timeline for a single DS trial as well as the differences between the two trial types are depicted in *Figure 1B*. Each session in this phase consisted of 30 discrete trials separated by a variable inter-trial interval – the start of each trial was signaled by the illumination of a discriminative stimulus for 30 s, following which rats were given access to the central retractable lever for 60 s. During this initial trial training, each session consisted of only one of two possible trial types – trials in which cocaine reward was available (DS+ trials) or trials in which cocaine reward was not available (DS- trials).

DS+ trials were signaled by the same DS used during continuous access self-administration (light on right side of lever, counterbalanced for red or white light). During DS+ trials, responses on the lever were rewarded under a fixed-ratio-1 (FR1) reinforcement schedule and cocaine reward at a unit dose of 0.75 mg/kg/infusion (0.1 ml/infusion) was delivered over 3.5 s. This infusion duration also served as the timeout period, during which lever presses were recorded but not reinforced. Additional lever presses during this 60 s period were also reinforced on the same schedule. Similar to self-administration training, delivery of cocaine in these trials was not paired with discrete cues. Sixty

seconds after lever presentation, the DS+ was turned off and the lever retracted, signaling the end of the trial.

DS- trials were signaled by the other available DS (light on left side of lever, counterbalanced for red or white light). During DS- trials, all responses on the lever were recorded but not reinforced. Sixty seconds after lever presentation, the DS- was turned off and the lever retracted, signaling the end of the trial.

All rats were trained on DS+ trials in the morning session and DS- trials in the afternoon session. We used two behavioral measures to monitor training during this phase – (*O'Brien, 2005*) the total number of DS+ vs. DS- trials with at least one lever press and (*Wikler, 1973*) the total number of responses made during DS+ vs. DS- trials during each 3 hr session.

### Discrimination training for cocaine reward

We trained rats on the trial-based discrimination procedure for two 3 hr sessions per day, separated by at least 30 min. In each of these sessions, rats received a total of 60 discrete trials; 30 DS+ trials and DS- trials were intermixed and presented in a pseudorandomized order such that rats received no more than two consecutive presentations of the same trial type during the session. Similar to the previous phase of training, we recorded (*O'Brien, 2005*) the total number of DS+ vs. DS- trials with at least one lever press and (*Wikler, 1973*) the total number of responses made during DS+ vs. DS-trials during each 3 hr session.

### Food self-administration

We trained male and female rats to lever press for palatable food reward (TestDiet, USA; Catalogue # 1811155, 12.7% fat, 66.7% carbohydrate, and 20.6% protein) during one 1-hr session per day. We gave rats the 45 mg food pellets in their home cage 1 day prior to the start of training and then used crushed food pellets when necessary to get rats to engage with the lever during initial continuous access training. The start of a session was signaled by the illumination of a light cue on the right of a central retractable lever followed 30 s later by the presentation of the retractable lever for 60 min. The light remained on for the duration of the session and served as a discriminative stimulus for palatable food reward availability. The same light was later used as a discriminative stimulus to signal availability of palatable food reward during trial-based discrimination training. Throughout the session, responses on this lever were rewarded under a fixed-ratio-1 (FR1) reinforcement schedule. Successful completion of the FR requirement led to the delivery of three 45 mg 'preferred' or palatable food pellets over 3.5 s. This reward delivery duration was not paired with any discrete cues and served as the timeout period, during which lever presses were recorded but not reinforced. At the end of the 1 hr session, the discriminative stimulus was turned off and the lever was retracted. We recorded (*O'Brien, 2005*) the total number of lever presses and (*Wikler, 1973*) the total number of rewards received during the entire session. We gave rats up to three training sessions to acquire stable self-administration responding before switching them to trial training for palatable food reward.

### Trial training for palatable food reward

We trained rats in two 1-hr trial training sessions in 2 days. We gave rats two trial training sessions before trial-based discrimination training to (*O'Brien, 2005*) habituate rats to the trial format and (*Wikler, 1973*) introduce the two possible trial contingencies separately before we mixed them together during discrimination sessions. The timeline for a single DS trial and the differences between the two trial types are depicted in *Figure 2B*. Each session in this phase consisted of 30 discrete trials separated by a variable inter-trial interval – the start of each trial was signaled by the illumination of a discriminative stimulus for 30 s, following which rats were given access to the central retractable lever for 60 s. During this initial trial training, each session consisted of only one of two possible trial types – trials in which palatable food-reward was available (DS+ trials) or trials where no palatable food reward was available (DS- trials).

DS+ trials were signaled by the same DS used during continuous access self-administration (light on right side of lever, counterbalanced for red or white light). During DS+ trials, responses on the lever were rewarded under a fixed-ratio-1 (FR1) reinforcement schedule. FR completion resulted the delivery of a single 45 mg palatable food pellet after 1 s and a 3.5 s timeout period during which lever presses were recorded but not reinforced. Additional lever presses during this 60 s period

were also reinforced on the same schedule. Similar to self-administration training, delivery of food-reward in these trials was not paired with discrete cues. Sixty seconds after lever presentation, the DS+ was turned off and the lever retracted, signaling the end of the trial.

DS- trials were signaled by the other available DS (light on left side of lever, counterbalanced for red or white light). During DS- trials, all responses on the lever were recorded but not reinforced. Sixty seconds after lever presentation, the DS- was turned off and the lever retracted, signaling the end of the trial.

All rats were trained on DS+ trials in the morning session and DS- trials in the afternoon session. We used two behavioral measures to monitor training during this phase – (*O'Brien, 2005*) the total number of DS+ vs. DS- trials with at least one lever press and (*Wikler, 1973*) the total number of responses made during DS+ vs. DS- trials during each 3 hr session.

## Discrimination training for palatable food reward
We then trained rats on the trial-based discrimination procedure for one 2 hr session per day. In each of these sessions, rats received a total of 60 discrete trials; 30 DS+ trials and DS- trials were intermixed and presented in a pseudorandomized order such that rats received no more than two consecutive presentations of the same trial type during the session. Similar to the previous phase of training, we recorded (*O'Brien, 2005*) the total number of DS+ vs. DS- trials with at least one lever press and (*Wikler, 1973*) the total number of responses made during DS+ vs. DS- trials during each 3 hr session.

## Abstinence phase
During the abstinence phase for both experiments, we housed rats in individual cages in the animal facility and handled them 1–2 times per week. In Experiment 1, after the rats successfully acquired discrimination, we housed them in the vivarium for up to 400 additional days and tested them repeatedly for relapse after progressively longer durations of abstinence from cocaine. In Experiment 2, after rats successfully acquired discrimination, we housed them in the vivarium for up to 200 additional days and tested them repeatedly for relapse after progressively longer durations of abstinence from palatable food reward.

## Relapse test
In Experiments 1 and 2, the experimental conditions during relapse tests were the same as the corresponding trial-based discrimination training session, except that responses on the lever were not reinforced in either DS+ or DS- trials (extinction conditions). In Experiment 1, infusion pumps were turned off during all relapse tests and all surviving rats were tested 1, 21, 60, 120, 200, 300, and, 400 days after the last discrimination training session. In Experiment 2, pellet dispensers were turned off during all relapse tests and all surviving rats were tested 1, 21, 60, 120, and, 200 days after the last discrimination training session. As with discrimination training, we recorded (*O'Brien, 2005*) the total number of DS+ vs. DS- trials with at least one lever press and (*Wikler, 1973*) the total number of responses made during DS+ vs. DS- trials during the entire relapse test session. We operationally define the term relapse as the continuation of non-reinforced discriminated drug seeking after a period of abstinence.

## Cocaine-primed reinstatement test
In Experiment 1, after the final relapse test (day 400), we tested the rats (n = 11) for cocaine-priming induced reinstatement during four separate sessions, run on consecutive days (days 401–404). On test days for cocaine-priming induced reinstatement, we gave the rats an intraperitoneal (IP) injection of saline or cocaine 10 min prior to the start of the test session. We chose an ascending dose order for cocaine in order (10, 20 mg/kg) to minimize a carry-over effect of a given priming dose on the subsequent priming dose. We tested the same rats for saline-primed reinstatement before and between cocaine-primed reinstatement tests. The experimental conditions during reinstatement test were the same as the trial-based discrimination training session, except that infusion pumps were turned off for the duration of the test and responses on the lever were not reinforced in either DS + or DS- trials (extinction conditions). As with discrimination training and relapse tests, we recorded (a) the total number of DS+ vs. DS- trials with at least one lever press and (b) the total number of

responses made during DS+ vs. DS- trials during the entire cocaine-primed reinstatement test session. The cocaine priming doses were based on previous studies using the reinstatement model (*Kalivas and McFarland, 2003*).

## Statistical analyses

As described earlier, not all rats completed all phases of the experiments. In Experiment 1, two of the 16 rats failed to complete discrimination training and were excluded from the study. Three of the 14 remaining test rats died during abstinence and did not complete all relapse tests. In Experiment 2, all 16 rats completed discrimination training and were tested repeatedly for relapse during abstinence. One test rat died during abstinence and did not complete all relapse tests. Therefore, we used maximum-likelihood-based multilevel models (SAS Proc Mixed) rather than ordinary-least-squares repeated-measures analyses of variance to account for missing data. Both approaches achieve the same objectives, but maximum-likelihood models obviate imputation of missing data and permit more accurate modeling of nonhomogeneity of variance across unevenly spaced time points.

We conducted all statistical analysis on two behavioral measures - (*O'Brien, 2005*) the total number of trials of each DS type with at least one lever press (denoted as '*successful*' *trials*) and (*Wikler, 1973*) the total number of responses made during each DS trial type over the entire session (denoted as *lever presses*). We followed up on statistically significant main effects or interactions with post-hoc tests as described below. Because some of our models yielded multiple main effects and interactions, we report only those that are critical for data interpretation. In preliminary analyses controlling for sex, we saw sex differences in the acquisition of discrimination, but not in the effects of interest (e.g. the intensity of potentiated seeking or time course of incubation of DS-controlled responding). Therefore, we collapsed our analyses across sex for both experiments. Sex-disaggregated data for all phases of each experiment are provided as figure supplements linked to the main figures (*Figure 1—figure supplement 1* and *Figure 2—figure supplement 1*) and statistical output for all analyses, including those controlling for sex are provided as associated source data files (*Figure 1—source data 1* and *Figure 2—source data 1*).

In Experiment 1, for the analysis of discrimination (*Figure 1C*, n = 14), we used a two-way factorial model with within-subject factors of discrimination training session (sessions 5–14) and DS type (DS+, DS-), accompanied by Tukey's Honest Significant Difference (HSD) test where appropriate for pairwise comparisons between DS+ and DS- for each training session. For the repeated relapse tests (*Figure 1D*, n = 11–14), we used mixed two-way factorial models with the within-subjects factors duration of forced abstinence (1, 21, 60, 120, 200, 300, and 400 days) and DS type (DS+, DS-), followed by Dunnett's test for pairwise comparisons between the day 1 relapse test and each of the following days' relapse tests. We also used Tukey's HSD for pairwise comparisons between DS+ and DS- for each relapse-test day.

For the tests of priming-induced reinstatement (*Figure 1E*, n = 11), we used 2-way models with the within-subjects factors cocaine priming dose (0, 10, 0, 20 mg/kg) and DS type (DS+, DS-). We used Tukey's HSD for pairwise comparisons between different cocaine priming doses within each DS trial type. We also used Tukey's HSD for pairwise comparisons between DS+ and DS- for each reinstatement-test day.

In Experiment 2, for the analysis of discrimination (*Figure 2C*, n = 16), we used a two-way factorial model with within-subject factors of discrimination training session (sessions 3–13) and DS type (DS+, DS-), accompanied by Tukey's HSD test for pairwise comparisons between DS+ and DS- for each training session. For the repeated relapse tests (*Figure 2D*, n = 15–16), we used two-way factorial models with the within-subjects factors duration of forced abstinence (1, 21, 60, 120, and 200 days) and DS type (DS+, DS-), followed by Dunnett's tests for pairwise comparisons between the day 1 relapse test and each of the following days' relapse testing. We also used Tukey's HSD for pairwise comparisons between DS+ and DS- on each relapse-test day.

In all models, we used a spatial-power error structure to account for autocorrelation across unevenly spaced intervals; this is similar to the use of a Huynh-Feldt or Greenhouse-Geisser correction in a repeated-measures ANOVA. Alpha (significance) level was set at 0.05, two-tailed.

## Acknowledgements

This research was conducted in compliance with the guidelines outlined in the Guide for the Care and Use of Laboratory Animals (8th edition; http://grants.nih.gov/grants/olaw/Guide-for-the-Care-and-Use-of-Laboratory-Animals.pdf) and was approved by the Institutional Animal Care and Use Committee and Institutional Biosafety Committee of the Intramural Research Program of the National Institute on Drug Abuse. We thank Drs. Leslie R Whitaker and Sam A Golden for their thoughtful comments during the writing of this manuscript.

## Additional information

### Funding

| Funder | Grant reference number | Author |
| --- | --- | --- |
| National Institute on Drug Abuse | DA000467-15 | Bruce T Hope |

The funders had no role in study design, data collection and interpretation, or the decision to submit the work for publication.

### Author contributions

Rajtarun Madangopal, Conceptualization, Data curation, Formal analysis, Supervision, Investigation, Visualization, Methodology, Writing—original draft, Writing—review and editing; Brendan J Tunstall, Conceptualization, Data curation, Formal analysis, Investigation, Methodology, Writing—original draft, Writing—review and editing; Lauren E Komer, Jennifer K Hoots, Data curation, Investigation, Writing—review and editing; Sophia J Weber, Data curation, Formal analysis, Investigation, Writing—review and editing; Veronica A Lennon, Data curation, Writing—review and editing; Jennifer M Bossert, Conceptualization, Resources, Supervision, Investigation, Methodology, Project administration, Writing—review and editing; David H Epstein, Formal analysis, Supervision, Writing—review and editing; Yavin Shaham, Conceptualization, Resources, Supervision, Methodology, Project administration, Writing—review and editing; Bruce T Hope, Conceptualization, Formal analysis, Supervision, Funding acquisition, Investigation, Methodology, Writing—original draft, Project administration, Writing—review and editing

### Author ORCIDs

Rajtarun Madangopal  http://orcid.org/0000-0001-6202-302X
Bruce T Hope  http://orcid.org/0000-0001-5804-7061

### Ethics

Animal experimentation: This study was performed in strict accordance with the recommendations in the Guide for the Care and Use of Laboratory Animals of the National Institutes of Health (8th edition; http://grants.nih.gov/grants/olaw/Guide-for-the-Care-and-Use-of-Laboratory-Animals.pdf). All rat experiments were approved by the Institutional Animal Care and Use Committee (Protocol# 17-BNRB-203) of the Intramural Research Program of the National Institute on Drug Abuse.

### Decision letter and Author response

Decision letter https://doi.org/10.7554/eLife.44427.014
Author response https://doi.org/10.7554/eLife.44427.015

## Additional files

### Supplementary files

• Transparent reporting form
DOI: https://doi.org/10.7554/eLife.44427.012

## Data availability

All data generated or analyzed during this study, and needed to evaluate the conclusions in the paper, are included in the manuscript and supplementary materials.

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
