## [Decision Letter]

Thank you for submitting your article "Discriminative stimuli are sufficient for incubation of cocaine craving" for consideration by *eLife*. Your article has been reviewed by three peer reviewers, one of whom is a member of our Board of Reviewing Editors, and the evaluation has been overseen by Laura Colgin as the Senior Editor. The following individuals involved in review of your submission have agreed to reveal their identity: Gavan McNally (Reviewer #2) and Jeff Grimm (Reviewer #3).

The reviewers have discussed the reviews with one another and the Reviewing Editor has drafted this decision to help you prepare a revised submission.

Summary:

The data in this report demonstrate incubation in the excitatory influence of a discriminative stimulus that signals cocaine ability over cocaine seeking behavior in abstinence. The data show that the ability of the DS+ to cause relapse-like behavior to cocaine seeking increases over 60 days of abstinence and maintains for 300 days. Conversely, the influence of a DS+ over food-seeking decreases over time, as would be expected by extinction. These data extend prior reports of incubation of craving induced by drug-paired conditional stimulus to a discriminative stimulus. This is important because it is often DS+'s that cause relapse in human drug abusers. A particular strength of the study is how long the relapse testing was carried out, demonstrating that the DS+ can control a rat's drug seeking for over half its lifetime, as well as the inclusion of both sexes.

Essential revisions:

You will see below that each reviewer raised a couple of moderate concerns. Please address these in your revision/rebuttal including:

1) Revision of the term CS-independent incubation.

2) Include discussion on the growth and maintenance of this DS+ incubation v. the incubation previously demonstrated with a cocaine CS+.

3) Move Supplementary Figure 1 to the main text.

4) Revise the rationale for the study in the Introduction according to reviewer #2's point 2.

5) Clarify what is meant by "in addition to setting the occasion for drug-seeking behavior, the DS+ can acquire excitatory motivational properties" (see reviewer 2 point #3).

6) Consider the possibility that other contextual conditions behaviors could show incubation (see reviewer 3 point #2).

7) Temper the conclusions regarding whether there could ever be incubation of a DS+ for food.

8) Report body weight and discuss whether this influenced the results of the food study with respect to sex differences.

*Reviewer #1:*

The data in this report demonstrate incubation in the excitatory influence of a discriminative stimulus that signals cocaine ability over cocaine seeking behavior in abstinence. The data show that the ability of the DS+ to cause relapse-like behavior to cocaine seeking increases over 60 days of abstinence and maintains for 300 days. Conversely, the influence of a DS+ over food-seeking decreases over time, as would be expected by extinction. These data extend prior reports of incubation of craving induced by drug-paired conditional stimulus to a discriminative stimulus. This is important because it is more often DS+'s than CS+'s that cause relapse in human drug abusers. A particular strength of the study is how long the relapse testing was carried out, demonstrating that the DS+ can control a rat's drug seeking for over half its lifetime. This is an excellent report that will likely be very beneficial to those looking to model relapse in rodents so as to better understand its neural correlates. I have a few moderate concerns.

1) My main concern is with the use of the term CS-independent incubation. This is misleading. The DS+ here both serves as a DS+, but likely also enters into a Pavlovian relationship with the cocaine US. Indeed, the DS+ serves as the most proximal and reliable salient predictor of cocaine delivery. Thus, although the programmed intent of the experimenters is for the stimulus to serve as a DS+, it is not possible to rule out that it is also becoming a predictive CS+ for the animal. This is important for understanding the cause (both psychological and neural) of the relapse. Does the animal fail to extinguish the DS+ and thus continues to think it signals cocaine availability even after evidence to the contrary? Or is the DS+ eliciting cocaine craving (much like a CS+ would) that triggers lever presses for cocaine. These are not mutually exclusive, and the answer might point to the neural circuitry involved. There are experimental ways to tease this out, but at the very least this should be discussed and the term CS-independent incubation should be revised.

2) It would have been ideal here to compare the DS+ incubation to CS+ incubation, in the absence of this, it would be helpful to provide some discussion on the growth and maintenance of this DS+ incubation v. the incubation that has been repeatedly demonstrated with a cocaine CS+.

*Reviewer #2:*

This is a very interesting and technically strong manuscript mapping the time course of incubation of responding for cocaine controlled by discriminative stimuli. There are several notable features. First, the use of a discrete trials procedure. This procedure lends itself very well to assessing behavioural specificity of any change in responding and indeed to future work with recording approaches. Second, the extensive timecourse over which animals were tested. The authors tested animals (cocaine at least) up to 400 days of forced abstinence. Third, the use of the same discrete trials procedure for food reward was helpful because it provides some confidence in comparing behavior controlled by stimuli for the two reinforcers. As such, I think this manuscript is an important contribution to the literature.

The manuscript is very well written and presented, the authors are also careful in their interpretation. The figures are very helpful. I don't have any further work to recommend but did have three comments.

1) Given that the authors have gone to the trouble of examining both sexes, I wondered if these data may get more value if they are in the manuscript proper, rather than the supplementary. These are heroic experiments, the data are meaningful, and it would be a shame if the data were 'lost' in the supplement.

2) I did not find some of the arguments in the Introduction justifying the need for a DS based study especially compelling. Although drug-associated Pavlovian CSs are trained due to their relationship with the drug US, they can be encountered in the 'real world' well separated from the interoceptive effects of the drug (e.g., an advertisement for a cigarette or alcoholic beverage). So, CSs, like DSs, can exert control in advance of drug taking. On the other hand, a justification that DSs are important for stimulus control of drug taking and little is known about how their properties change across abstinence – and certainly the kinds of intervals used here – is very compelling.

3) I struggled with the extra inference in the Discussion that "in addition to setting the occasion for drug-seeking behavior, the DS+ can acquire excitatory motivational properties". I am not really sure what is meant here (i.e. independently of the effect [incubation] that is being explained). By way of example, at least one account of occasion setting (Brandon and Wagner) supposes that occasion setting is based on emotive/motivational modulation. I hoped that the authors could unpack a little more what they mean here, especially compared with the loss of DS control in the food experiment.

*Reviewer #3:*

The manuscript from Madangopal and colleagues describes the results of an initial examination of incubation of cocaine craving in non-food-deprived male and female rats, specifically incubation of responding controlled by a discriminative stimulus.

In Experiment 1, rats rapidly acquired cocaine self-administration across 6 sessions (over 3 days) and then discriminated cocaine availability in a series of 90 trials over 7 days. Rats were then tested for DS+ and DS- appropriate responding in 3 h sessions 1, 21, 60, 120, 200, 300, and 400 days into abstinence from cocaine. In a subsequent test, responding was assessed following ascending doses of experimenter-delivered cocaine. DS+ responding incubated, peaking at 60 days post-abstinence with active lever responding remaining elevated at the 200-day mark. DS+ responding was reinstated by cocaine; this was not dose-dependent.

Experiment 2 followed essentially the same design as Experiment 1, but rats self-administered food pellets instead of cocaine and the final test day was day 200. As with cocaine, discrimination was obtained rapidly. Incubation was not observed, however. Responding decreased over 200 days; DS+/DS- discrimination was not observed after 60 days of abstinence.

The main finding is that cocaine-associated DS+/DS- controlled responding persists for nearly a year, and that it also incubates for a substantial portion of that year. Identifying the influence of a discriminative stimulus on protracted relapse behaviors, including incubation, is novel. This is an important finding.

1) The DS+ approach (e.g. good controls) used in the present study nicely demonstrates how DS+ and DS- can control reward-paired seeking behaviors.

2) Other contextual conditioned behaviors are likely not picked up in this and most other procedures-for example, the subjects likely are more energized over incubation as they are brought to, and then placed into, the operant chambers; this could be measured as increased locomotion in the operant chambers.

3) Clearly, incubation was not observed in the food study. However, it is premature to discount whether training with a non-drug reinforcer would produce DS+/DS- incubation. There are quantitative and qualitative differences to consider. For example, rate of responding in the discrimination leg of the food study differed from the cocaine study by several times. Also, while rats prefer the pellet used in this study over others, it is yet food-something they are provided ad libitum over the course of the study. Cocaine is only provided during training. It is possible that training with pure sugar, or a fat+sucrose reinforcer (e.g. Ensure) would have led to different results.

4) Body weight is not controlled for in the food pellet study. Females likely weighed much less than males yet self-administered pellets to a similar extent.

5) Just a curiosity for a potential follow-up: It appears that females responded at a higher rate with the cocaine DS+ at the 300- and 400-day time points. Do the authors consider that there is possibly a sex effect there, that is just missed (statistically speaking) due to the complexity of the ANOVA?

---

## [Author Response]

Essential revisions:You will see below that each reviewer raised a couple of moderate concerns. Please address these in your revision/rebuttal including:1) Revision of the term CS-independent incubation.

Thank you for your suggestion. We have edited the entire manuscript to omit the use of the term CS-independent incubation. We instead refer to the same as ‘DS-controlled incubation’ or ‘incubation in the absence of previously drug-paired CSs’.

2) Include discussion on the growth and maintenance of this DS+ incubation v. the incubation previously demonstrated with a cocaine CS+.

Thank you for your suggestion. We have revised the text in the Discussion to address this issue.

3) Move Supplementary Figure 1 to the main text.

Thank you for this suggestion. We have moved both supplementary figures and associated statistical summary tables to the main text. See Figure 1—figure supplement 2 and Figure 2—figure supplement 1. As per *eLife*’s guidelines, the figure supplements are now referred to in the legend of the associated primary figure and in the manuscript text where appropriate. Statistical analysis summary tables for the two experiments are now linked to the main figure as source data files.

4) Revise the rationale for the study in the Introduction according to reviewer #2's point 2.

Thank you for this suggestion. We agree with the reviewer’s comment and have omitted the following sentence from the Introduction:

‘From a translational perspective, this is problematic: when an abstinent addict encounters this kind of stimulus (e.g., the sensation of vapor inhaled into the lungs or the early interoceptive effects of the drug), relapse will have already happened.’

We have also edited the Abstract to highlight the importance of discriminative stimulus control over drug taking and drug seeking during abstinence.

5) Clarify what is meant by "in addition to setting the occasion for drug-seeking behavior, the DS+ can acquire excitatory motivational properties" (see reviewer 2 point #3).

Thank you for your feedback about the statement in the original submission and your comment regarding the occasion setting property of DSs and the strength of DS based stimulus control. We have revised the Abstract and added a paragraph to the Discussion to address this issue.

6) Consider the possibility that other contextual conditions behaviors could show incubation (see reviewer 3 point #2).

We have highlighted in both the Introduction (second paragraph) and Discussion (second paragraph) that in a previous study we found no evidence for incubation of the response to the drug-associated context.

We thank reviewer 3 for his suggestion regarding locomotor activity but unfortunately our operant boxes are not equipped with photobeams to measure locomotor activity so we could not test this idea. In Aoyama et al. (2014 Appetite 72:114-122), photobeam breaks were assessed during cocaine seeking. However, when measuring locomotion in a context where rats were previously trained to make an operant response for cocaine, it is difficult to disentangle conditioned locomotion from the effects of stimuli that enhance motivation for cocaine. We know that the motivation for cocaine incubates and is indexed by an increased behavioral output on the part of the rat (i.e., lever pressing) that is likely to indirectly increase measures of locomotion during the test. Furthermore, in conditions where the context was paired with cocaine in the absence of any operant response on the part of the animal, we did not observe incubation of locomotor behavior conditioned to a context (Hope et al. (2006) Eur. J. Neurosci. 24:867-875).

7) Temper the conclusions regarding whether there could ever be incubation of a DS+ for food.

We have modified the conclusions as recommended (Discussion, sixth paragraph).

8) Report body weight and discuss whether this influenced the results of the food study with respect to sex differences.

Thank you for your suggestion. We have included plots for subject body weight for each experiment (disaggregated by sex) as a figure supplement (Figure 1—figure supplement 2, and Figure2—figure supplement 3). We also include plots for number of pellets earned (denoted *Pellets*) and weight normalized food reward consumption (denoted as *Food reward*) during discrimination training (disaggregated by sex) as a figure supplement (Figure 2—figure supplement 2). We ran a 2-way ANOVA with between-subjects factor of Sex and within-subjects factor of Session to identify possible sex differences in either measure during training. Male and female rats consumed similar amounts of palatable food during training (no main effect of Sex for the *Pellets* measure) and this amounted to higher levels of food consumption per unit body weight (main effect of Sex for the *Food reward* measure) owing to differences in body weight between the two groups.

Reviewer #1:

[…] I have a few moderate concerns.1) My main concern is with the use of the term CS-independent incubation. This is misleading. The DS+ here both serves as a DS+, but likely also enters into a Pavlovian relationship with the cocaine US. Indeed, the DS+ serves as the most proximal and reliable salient predictor of cocaine delivery. Thus, although the programmed intent of the experimenters is for the stimulus to serve as a DS+, it is not possible to rule out that it is also becoming a predictive CS+ for the animal. This is important for understanding the cause (both psychological and neural) of the relapse. Does the animal fail to extinguish the DS+ and thus continues to think it signals cocaine availability even after evidence to the contrary? Or is the DS+ eliciting cocaine craving (much like a CS+ would) that triggers lever presses for cocaine. These are not mutually exclusive, and the answer might point to the neural circuitry involved. There are experimental ways to tease this out, but at the very least this should be discussed and the term CS-independent incubation should be revised.

Thank you for your suggestion about the term CS-independent incubation. We agree and have edited the entire manuscript to omit the use of the term CS-independent incubation. We instead refer to the same as ‘DS-controlled incubation’ or ‘incubation in the absence of previously drug-paired CSs’.

We also appreciate your comment regarding the possibility that the DS could become a predictive CS+ for the animal. We have added text to the revised Discussion to address this excellent comment.

2) It would have been ideal here to compare the DS+ incubation to CS+ incubation, in the absence of this, it would be helpful to provide some discussion on the growth and maintenance of this DS+ incubation v. the incubation that has been repeatedly demonstrated with a cocaine CS+.

Thank you for your suggestion. We have addressed this issue in the revised Discussion.

Reviewer #2:

[…] The manuscript is very well written and presented, the authors are also careful in their interpretation. The figures are very helpful. I don't have any further work to recommend but did have three comments.1) Given that the authors have gone to the trouble of examining both sexes, I wondered if these data may get more value if they are in the manuscript proper, rather than the supplementary. These are heroic experiments, the data are meaningful, and it would be a shame if the data were 'lost' in the supplement.

Thank you for this suggestion and the positive comments. Please see our response to Editor comment #3.

2) I did not find some of the arguments in the Introduction justifying the need for a DS based study especially compelling. Although drug-associated Pavlovian CSs are trained due to their relationship with the drug US, they can be encountered in the 'real world' well separated from the interoceptive effects of the drug (e.g., an advertisement for a cigarette or alcoholic beverage). So, CSs, like DSs, can exert control in advance of drug taking. On the other hand, a justification that DSs are important for stimulus control of drug taking and little is known about how their properties change across abstinence – and certainly the kinds of intervals used here – is very compelling.

Thank you for your positive feedback. We agree with your comments and have omitted the following sentence from the Introduction:

‘From a translational perspective, this is problematic: when an abstinent addict encounters this kind of stimulus (e.g., the sensation of vapor inhaled into the lungs or the early interoceptive effects of the drug), relapse will have already happened.’

We have also edited the Abstract to highlight the importance of discriminative stimulus control over drug taking and drug seeking during abstinence

3) I struggled with the extra inference in the Discussion that "in addition to setting the occasion for drug-seeking behavior, the DS+ can acquire excitatory motivational properties". I am not really sure what is meant here (i.e. independently of the effect [incubation] that is being explained). By way of example, at least one account of occasion setting (Brandon and Wagner) supposes that occasion setting is based on emotive/motivational modulation. I hoped that the authors could unpack a little more what they mean here, especially compared with the loss of DS control in the food experiment.

Thank you for your feedback about the statement in the original submission and your comment regarding the occasion setting property of DSs and the strength of DS based stimulus control. We have revised the Discussion to address your concerns.

We have also expanded the Discussion about differences between incubation controlled by drug- and food- DSs.

Reviewer #3:

The manuscript from Madangopal and colleagues describes the results of an initial […] 1) The DS+ approach (e.g. good controls) used in the present study nicely demonstrates how DS+ and DS- can control reward-paired seeking behaviors.

Thank you for your positive comment.

2) Other contextual conditioned behaviors are likely not picked up in this and most other procedures-for example, the subjects likely are more energized over incubation as they are brought to, and then placed into, the operant chambers; this could be measured as increased locomotion in the operant chambers.

Thank you for this suggestion but unfortunately our operant boxes are not equipped with photobeams to measure locomotor activity so we could not test this idea. In Aoyama et al. (2014 Appetite 72:114-122), photobeam breaks were assessed during cocaine seeking. However, when measuring locomotion in a context where rats were previously trained to make an operant response for cocaine, it is difficult to disentangle conditioned locomotion from the effects of stimuli that enhance motivation for cocaine. We know that the motivation for cocaine incubates and is indexed by an increased behavioral output on the part of the rat (i.e., lever pressing) that is likely to indirectly increase measures of locomotion during the test. Furthermore, in conditions where the context was paired with cocaine in the absence of any operant response on the part of the animal, we did not observe incubation of locomotor behavior conditioned to a context (Hope et al. (2006) Eur. J. Neurosci. 24:867-875).

3) Clearly, incubation was not observed in the food study. However, it is premature to discount whether training with a non-drug reinforcer would produce DS+/DS- incubation. There are quantitative and qualitative differences to consider. For example, rate of responding in the discrimination leg of the food study differed from the cocaine study by several times. Also, while rats prefer the pellet used in this study over others, it is yet food-something they are provided ad libitum over the course of the study. Cocaine is only provided during training. It is possible that training with pure sugar, or a fat+sucrose reinforcer (e.g. Ensure) would have led to different results.

Thank you for your suggestion. We addressed this issue in the revised Discussion.

4) Body weight is not controlled for in the food pellet study. Females likely weighed much less than males yet self-administered pellets to a similar extent.

Thank you for your comment. Please see our response to Editor comment #8.

5) Just a curiosity for a potential follow-up: It appears that females responded at a higher rate with the cocaine DS+ at the 300- and 400-day time points. Do the authors consider that there is possibly a sex effect there, that is just missed (statistically speaking) due to the complexity of the ANOVA?

Thank you for this observation. Unfortunately, only 5 of 7 male rats survived at these timepoints. We used maximum-likelihood-based multilevel models (SAS Proc Mixed) rather than ordinary-least-squares repeated-measures analyses of variance to account for the missing data While there is a hint at a sex effect at later timepoints in the cocaine experiment, we did not see a 3-way interaction. We cannot rule out the possibility that increasing the group size might have produced a significant effect. To make the analyses more accessible we have moved the statistical output tables for both experiments to the main manuscript as described in our response to Editor comment #3.